# Studying Heterotypic Cell–Cell Interactions in the Human Brain Using Pluripotent Stem Cell Models for Neurodegeneration

**DOI:** 10.3390/cells8040299

**Published:** 2019-04-01

**Authors:** Liqing Song, Yuanwei Yan, Mark Marzano, Yan Li

**Affiliations:** Department of Chemical and Biomedical Engineering, FAMU-FSU College of Engineering, Florida State University, 2525 Pottsdamer St., Tallahassee, FL 32310, USA; ls13f@my.fsu.edu (L.S.); yyan73@wisc.edu (Y.Y.); mcm12g@my.fsu.edu (M.M.)

**Keywords:** pluripotent stem cells, heterotypic, microglia, endothelial, neural-vascular interactions, mesenchymal stem cells

## Abstract

Human cerebral organoids derived from induced pluripotent stem cells (iPSCs) provide novel tools for recapitulating the cytoarchitecture of the human brain and for studying biological mechanisms of neurological disorders. However, the heterotypic interactions of neurovascular units, composed of neurons, pericytes (i.e., the tissue resident mesenchymal stromal cells), astrocytes, and brain microvascular endothelial cells, in brain-like tissues are less investigated. In addition, most cortical organoids lack a microglia component, the resident immune cells in the brain. Impairment of the blood-brain barrier caused by improper crosstalk between neural cells and vascular cells is associated with many neurodegenerative disorders. Mesenchymal stem cells (MSCs), with a phenotype overlapping with pericytes, have promotion effects on neurogenesis and angiogenesis, which are mainly attributed to secreted growth factors and extracellular matrices. As the innate macrophages of the central nervous system, microglia regulate neuronal activities and promote neuronal differentiation by secreting neurotrophic factors and pro-/anti-inflammatory molecules. Neuronal-microglia interactions mediated by chemokines signaling can be modulated in vitro for recapitulating microglial activities during neurodegenerative disease progression. In this review, we discussed the cellular interactions and the physiological roles of neural cells with other cell types including endothelial cells and microglia based on iPSC models. The therapeutic roles of MSCs in treating neural degeneration and pathological roles of microglia in neurodegenerative disease progression were also discussed.

## 1. Introduction

Recent human-induced pluripotent stem cell (hiPSC) technology has provided a remarkable alternative for human in vitro models through the derivation of lineage-specific cells from patients with diverse neurological diseases. Brain organoids, which display 3D brain-like structure and function, have been generated from hiPSCs [1,2,3,4]. A recent 3D cortical organoid model formed using dual SMAD inhibition and fibroblast growth factor (FGF) signaling achieved cortical layer structure, which has potential to provide an innovative platform for pharmacological applications [5,6,7]. Some novel brain organoid models have been used for studying neurological disease progression [8], psychiatric disorders [9,10,11], and Zika virus infections [3,12,13,14,15].

However, human brain development is mediated by complex intercellular interactions between functionally different cell types including neurons, glia, endothelial cells, pericytes, and microglia (Figure 1). Significant knowledge gaps still exist due to the lack of good models to study interactions of neural cells with other cell types. For example, for neurotoxicity in vitro, toxins might target on distinct cell types or cellular interactions, which requires the culture models of multiple cell types. A recent study reported a generalized method to form organ buds of different types of tissues (e.g., liver, kidney, brain etc.) in vivo using mesenchymal stem cells (MSCs), endothelial cells, and hiPSC-derived tissue progenitor cells [16]. In addition, organ buds self-organized into multiple functional and vascularized organs such as liver or kidney after the transplantation [16,17]. This approach simply mixed the cells of different types and the cell organization and tissue structure are uncontrolled. In addition, the endothelial cells are not organ-specific. Recently, complex 3D organoids have been constructed recently from hiPSCs, such as cortical or cerebral organoids [4,10,18,19], to predict neurotoxicity of different drugs [20] and to use for an in vivo study [2]. By promoting the secretion of endogenous growth factors and extracellular matrix (ECM) proteins, the formation of heterogenic 3D tissue architecture recapitulates essential intercellular microenvironment and cell–matrix interactions of functional tissues [21,22]. However, the precise role of 3D intercellular organization and structural environment on the biological function of the organoids has not been well explored. 

In the context of 3D cultures and spheroids/organoids, MSC secretome is the modulators of neurogenic niche and promotes neural differentiation through trophic effects [23,24]. MSCs are also able to form spheroids and secrete anti-apoptotic and anti-inflammatory factors including prostaglandin E2 (PGE2) [25]. MSCs displayed higher homing ability to the injuries sites for neural protection, due to the increased expression of homing factor CXCR4 [26]. There are some phenotype overlaps for MSCs and brain specific pericytes, i.e., the tissue resident mesenchymal stromal cells [27,28,29]. Brain-specific pericyte-like cells can be induced through mesoderm or neural crest differentiation from hiPSCs [30].

Neural-vascular interactions, known as neural-vascular unit (NVU), play an important role in brain structure and function [31]. It has been suggested that organ-specific endothelial cells (ECs) secrete a unique set of growth factors that regulate tissue morphogenesis into desired tissue types [32,33]. Vascular cells can form spheroids to assemble blood vessels or as building blocks for scaffold-free tissue fabrication [33,34,35]. In vitro vascularization of organoids has been attempted for cardiac organoids, showing the enhanced cardiac cell function [36]. In vivo vascularization of organoids was demonstrated for the hiPSC-derived organ buds, in which the mixed hiPSC-derived progenitors and endothelial cells efficiently self-organize into functional and vascularized liver or kidney [16,17]. In particular, the blood–brain barrier (BBB) is involved in various neurological diseases development, drug administration, and nutrient transport [31,37]. Functional BBB models require the interactions of brain microvascular endothelial cells (BMECs), astrocytes, neurons, microglia, pericytes etc., which can be achieved using hiPSC-derived cells [38,39,40,41]. Thus, the rationale for the incorporation of BMECs and MSCs within brain organoids is to enable the formation of a pro-neurogenic niche that promotes brain tissue patterning and maturation.

Microglia are resident macrophage-like cells in the human brain [42,43,44] and they have two main functions: (1) immune defense; and (2) the maintenance and development of the central nervous system (CNS) [42,43]. Microglia restrain potential damage to the CNS and support tissue repair and remodeling, under the pro-inflammatory or anti-inflammatory conditions. They can phagocytose pathogens and cell debris, removing toxic molecules and protein deposits [45]. Moreover, microglia have been demonstrated to control synaptic plasticity through pruning during the development [43,46]. Microglia dysfunction has been found for various diseases such as Alzheimer’s disease, Parkinson’s disease, schizophrenia, Huntington’s disease, and brain tumors [47,48]. However, most of recent brain organoids from hiPSCs lack several critical components (e.g., vascular cells, BBB feature) in the human brain, one of which is the microglia [1], so fully mimicking the function of the human brain has not been achieved. 

The inefficient neural function of cells or culture models derived from hiPSCs motivates recent investigations of complex heterotypic interactions of different cell types to recapitulate cellular behaviors in vivo [1,49]. Our lab constructed 3D self-assembled hybrid spheroids by incorporating mesenchymal stem cells into hiPSC-derived neural progenitor cell (NPC) spheroids, which provide a promising platform for disease modeling as well as a transplantation entity to regenerate functional organs. Our study showed that MSCs promoted dorsal cortical spheroid formation [50]. Moreover, our lab also investigated the vascularization of cortical organoids in vitro through tri-culture of iNPCs, iECs, and human MSCs and investigated heterotypic neural-vascular-mesenchymal interactions (unpublished data). Built on these observations, it was suggested that incorporation of microglia-like cells may enhance neural protection ability of brain region-specific organoids. In order to incorporate microglia-like cells into the forebrain organoids, we have: (1) generated and characterized microglia-like cells from hiPSCs; (2) investigated the impacts of microglia-like cells on the microenvironment of 3D dorsal or ventral cortical organoids; and (3) investigated the impacts of microglia on neural protection and attenuating neural inflammation (unpublished data). These studies should advance our understanding of the effects of heterotypic cell–cell interactions on neural tissue function and establish a transformative approach to modulate cellular microenvironment during neurogenesis toward the goal of treating various neurological disorders.

## 2. The Physiological Role of Vascular Cells in the Central Nervous System 

### 2.1. Neurovascular Interactions

Neurovascular interactions are crucial for normal development of the central nervous system and proper brain functions [51]. In vivo, angiogenesis after neurological disorders promotes endogenous repair and supports axonal outgrowth [52]. Neurovascular dysfunction contributes to a variety of CNS disorders including ischemic stroke and Alzheimer’s disease (AD). For example, in AD, the neovascularization was inhibited by the accumulation of amyloid β (Aβ), resulting in the decoupling of neurovascular interactions and a loss of brain tissue homeostasis [53]. Recently, more interest has arisen in understanding the signaling between the vascular cells and the neural cells which can regulate the interactions between peripheral circulation and brain activity. In vitro models of the neurovascular environment are essential for the development of effective therapeutics and the understanding of disease pathology. The cross-talk between the neural cells and the vascular cells has been investigated through neural-vascular co-culture either in direct contract [54] or separated by a porous membrane in transwell systems [55,56]. Endothelial cells have been reported to secrete paracrine factors that stimulate the self-renewal and differentiation of neural stem cells via a Notch/Hes1 pathway in a co-culture study [55]. Previous indirect co-culture systems reported the delayed differentiation of neural progenitors implied by co-culturing with BMECs [57]. However, neuronal differentiation was promoted through the establishment of neurovascular niche and the direct association of vascular cells with neural cells, which is required for the capillary-like structure formation [55,56,58]. In addition, the establishment of neurovascular interactions in vitro promoted the functionality and maturation of cortical neurons (Figure 1) [54].

As to underlying mechanisms, robust tube formation and stabilization was mediated by vascular endothelial growth factor (VEGF) and brain-derived neurotrophic factor (BDNF) through neural progenitor-secreted nitric oxide (Figure 2) [59]. Moreover, the increased VEGF and BDNF may induce the self-renewal and differentiation of NPCs in a paracrine manner. Indeed, VEGF can act as a neurotrophic factor and the guidance cue for neural progenitors [54,60,61,62]. VEGF overexpression in brain significantly enhanced cortical newborn neurons and improved their differentiation into GABAergic neurons and the development of dendrites in rat brain after cerebral ischemia [61]. Under normal conditions, co-culturing with endothelial cells promoted synapse formation of cortical neurons, which was suggested to be mediated by VEGF/Flk-1/p38 and mitogen-activated protein kinase (MAPK) signaling [54].

Recent studies about in vitro 3D spheroid models of neurogenic niche involving direct cell–cell signaling showed that NPC survival and neuronal differentiation could be improved by 3D co-culturing with endothelial cells [2,16]. For instance, endothelial cells within the 3D cortical spheroids assembled into lumenized capillary-like structures, which have the applications for screening compounds for neurovascular dysfunction-related diseases, such as ischemic stroke [58]. Notch signaling activated by VEGF has been suggested as the underlying mechanism governing the neurovascular interactions, neurogenesis and angiogenesis [63]. The biological function of neurons co-cultured with endothelial cells varies within different culture conditions (i.e., VEGFR-2 signaling is active in ECs with high Notch activity, and VEGFR-3 drives EC growth with low Notch signaling activity), which activated different signaling pathways (VEGFR-2, VEGFR-3, or Notch pathways). These observations suggest that the establishment of functional neuronal network requires complex cell–cell and cell–matrix interactions rather than the secreted factors only to recapitulate the brain tissue, which contains neurons, microglia, endothelial cells, and glia [64].

### 2.2. The Blood–Brain Barrier (BBB) Models

The blood–brain barrier provides one of the best examples for how the neuronal-vascular interactions function to ensure a homeostatic environment for proper brain function [37]. For instance, the BBB can limit the access of molecules and peripheral cells to brain. Together with the basement membrane, endothelial cells, neural progenitor cells, astrocytes, and pericytes surround the central BBB core, altogether make the neurovascular unit (Figure 3). Those physical cellular interactions formed from multiple cell types contribute to the unique tightness and selective permeability of the BBB. Specifically, the BBB is an important barrier which allows for the diffusion of nutrients assisted by the distinct nutrient transporters and prevents the neurotoxic proteins into the brain with the enrichment of polarized efflux transporters. It has been shown that the impairment of BBB permeability is associated with neurodegenerative disorders, such as vascular dementia, stroke, amyotrophic lateral sclerosis, AD, and Parkinson’s disease. Considering that therapeutic drugs need to be delivered into the brain to treat BBB-related disorders, it is informative to investigate the permeability of novel drugs through the barrier before clinical trials. Taken together, in vitro modeling of the BBB is important to investigate the underlying cellular mechanisms of neurodegenerative disorders and serves as a powerful tool for pre-clinical drug permeability testing.

Current in vitro BBB models usually use BMECs from various non-human species, such as pig, rat, cow, monkey or mouse due to the limited availability of human primary BMECs, which restricts the application of these models in human because of species differences [40]. In particular, the differences in the expression and functionality of important BBB transporters such as P-glycoprotein is species-dependent. In addition, traditional human BBB models with a monoculture of BMECs lack the complexity of cellular interactions which lead to the lower barrier tightness, indicated by low transendothelial electrical resistance (TEER) compared to the BBB in vivo.

Human brain microvessels isolated from brain specimens or BMECs derived from hiPSCs have been used to establish in vitro BBB models (Table 1) [38,39,65,66]. Based on the procedure developed by Lippmann et al for differentiating BMECs from hiPSCs [67], Katt et al reported a modified protocol and generated the brain endothelial cells with BBB phenotype [68,69]. The expression of tight junction proteins, claudin-5, ZO-1, VE-cadherin, and occludin, and transporters, PGP, BCRP, MRP1, GLUT-1 and LAT-1 indicated the efficiency for derivation of BMECs. Moreover, the addition of retinoic acid was found to result in the unregulated VE-cadherin and good barrier integrity with TEER value up to 2,000 Ωcm^2^ [70]. Subsequently, one recent research study employed this in vitro BBB system in the characterization of permeability of cancer-targeting fluorescent agent, alkylphosphocholine (APC) [71]. Three different APC analogs, CLR1404, CLR1501, and CLR1502 were tested for permeability across a BBB model. It is known that lipophilicity and molecular weight are the determinants for the BBB permeability. Increased lipophilicity leads to the enhanced BBB permeability, while higher molecular weight reduces the BBB permeability. Diazepam was used as a highly BBB-permeable control and sucrose as a BBB-impermeable control. The transport of CLR1404 was basically driven by diffusion through the BBB, while the active efflux also played a role in the transport of CLR1501 and CLR1502. The underlying transport mechanisms of these small molecules showed significance in improving the outcomes of brain tumor patients in multimodality clinical management such as diagnostic imaging.

However, these monoculture models still have limitations, including low availability, low reproducibility, and low barrier tightness. These limitations of the current BBB models have motivated the researchers to find novel methods to recapitulate the low permeability of BBB in vivo. Thus, several co-culture models have been developed, including the models with all BBB components, i.e., ECs, pericytes, neurons, and astrocytes, derived from human PSCs [38,39,40,41].

### 2.3. In Vitro Co-Culture Modeling of BBB Using Human-Induced Pluripotent Stem Cells (hiPSCs)

The BBB is a layer of BMECs connected with neurons, astrocytes, pericytes, and microglia. NPCs prime the ECs toward a brain identity [40,49]. Astrocytes contribute to the BBB tightness and are characterized by intermediate filament protein glial fibrillary acidic protein (GFAP) and glial-specific calcium-binding protein B (S100β). Pericytes, directly contact with BMECs, are a heterogeneous MSC-like population in terms of morphology and marker expression, and contribute to BBB permeability and neural inflammation [81]. Human primary pericytes highly express platelet-derived growth factor receptor-beta (PDGFRβ) and alpha smooth muscle actin (αSMA). The absence of pericytes results in the low barrier tightness or the increase of BBB permeability. Co-culturing BMECs with astrocytes and pericytes enhances the barrier tightness by secreting factors and/or physical cellular interactions. 

HiPSCs are the ideal cell source for BBB modeling due to: (1) the unlimited availability of hiPSCs (self-renewal ability); and (2) the capability of patient-derived hiPSCs for the investigations of the underlying cellular mechanisms related to neurodegenerative diseases. Recently, a number of in vitro multicellular BBB models have been developed to investigate the crucial role of BBB in brain development, function, and disease progression (Table 1) [38,39,65,66,74]. Systematic comparison of different multicellular models showed the beneficial effects of pericytes on BBB tightness [38]. The triple culture of hiPSC-ECs, hiPSC-NPCs, and pericytes, or quadruple culture of hiPSC-ECs, hiPSC-NPCs, astrocytes, and pericytes, achieved the BBB characteristics with TEER value up to 2500 Ωcm^2^, which is comparable to the physiological TEER value in humans up to 5000 Ωcm^2^. In addition, the robust upregulation of BBB characteristic genes was observed in a quadruple culture model compared to hiPSC-ECs. However, the primary human brain pericytes and astrocytes were used, and hence the model is limited in scale and accessibility. Therefore, the development of isogenic BBB models through differentiation for hiPSC-derived astrocytes and pericytes is necessary. 

One recent isogenic BBB model with three cell types (BMECs, astrocytes, and neurons) was established based on the differentiation of BMECs from hiPSCs [39]. This study established differentiation protocols for hiPSC-derived astrocytes using 3D EZ spheres, the stable and expandable PSC-derived neural stem cell-like aggregates maintained by FGF-2 and epidermal growth factor (EGF) [82,83]. After the systematic comparison of different co-culture models with a monoculture of BMECs derived from hiPSCs, it was found that 3D EZ sphere-derived neurons and astrocytes at a ratio of 1:3 enhanced the barrier tightness of BMECs with the elevated TEER value around 886 Ωcm^2^. The examination of tight junction proteins, occludin and claudin-5, suggested that the enhanced barrier properties may be due to the improved formation and maintenance of tight junctions. This isogenic BBB model has potential applications in understanding cellular interplay of neurovascular units in diseased humans.

A novel human BBB micro-physiological system consisting of a 3D printed electrospun poly(lactic-co-glycolic) acid nanofibrous mesh, and a bilayer co-culture of human astrocytes and endothelial cells derived from hiPSCs [66]. This human BBB model showed high barrier integrity with tight junction protein expression, an effective permeability to sodium fluorescein, and higher trans-TEER value comparing to electrospun mesh-based counterparts. The co-culture of hiPSC-derived astrocytes and endothelial cells increased the tight junction protein expression and the TEER value. To increase the complexity of cell–cell interactions in BBB, spheroids of six cell types (including human primary astrocytes, endothelial cells, pericytes, and hiPSC-derived neuron stem cells, microglia, and oligodendrocyte progenitor cells) were formed, which expressed GLUT1, PGP, and ZO-1, the typical BBB markers [20]. The model responded to inorganic mercury toxicity and the neuro-toxicity to MPTP, a small molecule that can induce Parkinson’s disease. However, determination of TEER values for BBB in 3D spheroid models is still challenging.

One 3D BBB model was reported recently, with hiPSC-derived BMECs within templated type I collagen channels that mimic the cylindrical geometry, cell-ECM interactions, and shear flow typical for human brain post-capillary venules [80,84]. The microvessels recapitulated physiologically low solute permeability and quiescent EC behavior, with TEER value up to 2260 Ωcm^2^. The BBB model responded to inflammatory cytokines (e.g., TNF-α) via the upregulation of surface adhesion molecules (e.g., VCAM-1 and ICAM-1) and the increased leukocyte adhesion, but no changes in permeability. Another 3D BBB model used fibrin gel to promote blood vessel formation and the microfluidics to mimic blood flow [65]. Our study assembled 3D hiPSC-derived cortical spheroids/organoids with isogenic vascular spheroids in the presence of human MSCs (unpublished data). The incorporation of MSCs upregulated the secretion levels of cytokines VEGF-A, PGE2, and transforming growth factor (TGF)-β1. In addition, tri-cultured spheroids had high levels of TBR1 (deep cortical layer VI) and Nkx2.1 (ventral forebrain cells), and matrix remodeling genes, matrix metalloproteinase (MMP) 2 and MMP3, as well as Notch-1, indicating the crucial role of matrix remodeling and cell–cell communications on cortical organoid patterning. Moreover, the tri-culture system elevated BBB gene expression (e.g., GLUT-1), CD31, and tight junction protein ZO-1 expression. This forebrain-like model has potential applications in understanding heterotypic cell-cell interactions in diseased human brains and for screening novel drugs.

## 3. The Role of Mesenchymal Stem Cells (MSCs) on Neurodegeneration

There are considerable interests in the therapeutic potentials of human bone marrow derived MSCs (hMSCs), the pericyte-like cells [27,28]. Accumulating evidences suggest the therapeutic benefits of hMSCs on tissue repair are mainly attributed to paracrine signaling or cell-to-cell contacts instead of cell engraftment and differentiation at the injured site [85]. Previous reports suggest that the aggregation of hMSCs into 3D spheroids increases the therapeutic benefits by increasing the expression of CXCR4 to promote cell adhesion to endothelial cells, promoting anti-tumorigenic factor, IL-24 (a tumor suppressor protein [86]), and the enhanced anti-inflammatory properties [87]. As suggested, the properties of hMSC spheroids highly depend on the culture conditions [25]. Bartosh TJ et al reported a hanging drop protocol to generate MSC spheroids which had optimal levels of TNF-α stimulated gene/protein 6 (TSG-6) and stanniocalcin-1, as well as an anti-inflammatory/anti-apoptotic protein IL-24 [88,89]. Conditioned medium collected from 3D MSC spheroids attenuated the pro-inflammatory cytokines secretion, e.g. TNF-α, CXCL-2, and IL-6, from lipopolysaccharide (LPS)-stimulated macrophages and increased the secretion of anti-inflammatory cytokines including IL-10 and IL-1ra (Figure 4). In addition, it was found that the elevated PGE2 in conditioned medium of 3D hMSC spheroids played the key role in changing the stimulated macrophages from a pro-inflammatory phenotype to an anti-inflammatory phenotype. The secretion of PGE-2 was dependent on the activity of caspase and NF-кB activation [87,88]. Taken together, the hMSC spheroids may be more advantageous than the hMSC monolayer in modulating inflammatory reactions. 

Human MSC secretome has been demonstrated to promote the engraftment and neurogenesis of human NPCs by stimulating endogenous ECM secretion from NPC aggregates and enhancing cell proliferation [50,90]. These studies revealed that hMSC secretome including FGF-2, BDNF, and TGF-β1 were involved in the neurogenesis of NPCs. Indeed, inhibition of FGF-2, BDNF, and TGF-β1 signaling differentially reduced the cell adhesion, neurite extension and migration length of the replated NPC aggregates. In another study, Wharton’s Jelly-derived MSCs showed neuroprotective efficacy by reducing apoptosis of cortical neurons in an oxygen- and glucose-deprivation culture via a paracrine mechanism [91]. It is believed that the activation of Notch signaling may contribute to neuroprotective ability [92,93]. Co-culturing MSCs with adult NPCs enhanced cell proliferation and promoted neural differentiation by upregulating the Notch signaling receptor, Notch-1, and the downstream effector Hes1. Indeed, a significant increase of Notch-1 was observed in NPCs promoted by the conditioned medium from hMSC spheroids cultured in bioreactors, which is correlated with the enhanced neural differentiation. In addition, 3D spheroids of hMSCs showed the upregulated secretion of ECM remodeling proteins, MMP2 and MMP9 [25,50]. Chemokines induced by the injured cells could attract the migration of newborn NPCs. Previous studies showed that the chemotaxis function of VEGF and stromal derived factor (SDF)-1 on migration of NPCs was mediated by the upregulated endogenous MMP3 and MMP9 [94]. In addition to migration, the neurogenic effects of chemokines on NPCs was abolished by the blocking of MMP3 and MMP9, suggesting that the chemokines such as SDF-1 and VEGF may promote neural differentiation partially due to the elevated expression of MMPs. In another study, adult hippocampal NPCs overexpressed with MMP1 showed the enhanced proliferation and neuronal differentiation compared to control NPCs. The inhibition of NF-кB signaling abolished the enhanced proliferation and differentiation, suggesting that the MMP1 may play a role in neurogenesis of hippocampal NPCs by activation of NF-кB signaling [95]. 

In terms of angiogenesis, MSCs induced endothelial cell migration in a co-culture system and the conditioned medium of MSCs induced microvasculature formation of endothelial cells on Matrigel [91,96]. For the mechanisms, it is believed that MSC secretome contains anti-apoptotic and angiogenic factors, such as IL-6 and VEGF, via activation of the PI3K/Akt pathway in hypoxic endothelial cells to inhibit the apoptosis and stimulate angiogenesis. Taken together, MSC secretome promotes the neurogenesis and angiogenesis primarily through the secreted growth factors and the upregulated ECM remodeling.

## 4. The Physiological Role of Microglia in the Human Brain

### 4.1. Microglial Phenotype 

As the innate macrophages of CNS, microglia have shown the capability of phagocytosis of pathogens and cell debris during neural tissue development [97]. In addition, several studies have suggested the roles of microglia in regulating the neuronal activities and promoting neuronal differentiation is to act by secreting neurotrophic factors (e.g., BDNF and VEGF) and anti-inflammatory molecules (e.g., IL-10, TGF-β1, PGE2, etc.) [98,99,100]. 

Microglia emerge from Tie2+ expressing erythromyeloid progenitors from the embryonic yolk sac and later invade into the brain during embryonic development to transform into the resident ‘surveying microglia’ (Figure 5) [101]. The resident ‘surveying microglia’ adopt a classical ramified morphology under healthy condition. While under neurodegenerative conditions, microglia become activated and microglial morphologies change dramatically. In the meantime, the bone marrow-derived myeloid cells may penetrate the compromised blood–brain barrier and differentiate into microglia to infiltrate the CNS, which might be the second origin of microglia [102]. However, it was also suggested that infiltrating peripheral macrophages during pathologic conditions can reach the brain and increase the pool of active phagocytic cells in the brain, together with microglia, but they maintain their own identity and do not simply become microglia [103]. The ubiquitous distribution of microglia in the brain makes them the central cellular component of CNS inflammation, which is characterized by microglial activation. The ‘surveying microglia’ become activated under various environmental changes by functional receptor activation. For instance, microglia can sense neuronal damage through activation of purinergic P2 × 7 receptors (P2RY12) and CXCR3 on microglia surface via adenosine di-phosphate (ADP) and CXCL10 respectively from degenerating neurons [104,105]. However, the activated microglia can be both neurodegenerative and neuroprotective depending on the activation pathway (Figure 5). Surveying microglia can be classically activated (M1 phenotype) induced by LPS or IFN-γ to release pro-inflammatory molecules, such as TNF-α, IL-6, IL-1β, MMP, reactive oxygen species (ROS), and glutamate that cause neuronal damage [100,106]. Alternatively, microglia can be activated (M2 phenotype) by IL-4 or IL-13 to phagocytose pathogens and cell debris to induce an anti-inflammatory response with upregulated IL-10 and arginase 1. 

While macrophage/microglia polarization states have been defined as classical activation (M1), alternative activation (M2a), type II alternative activation (M2b) or acquired deactivation (M2c) [107], this strict and simplistic classification is not considered to adequately describe the complex physiology of microglial cells [108]. As a recent trend, the terms “homeostatic microglia” and “disease-associated microglia” based on molecular and protein signatures were proposed to provide insights into how these cells are regulated in healthy and diseased conditions and how they contribute to the maintenance of the neural environment [109]. Microglia display heterogeneity and plasticity, which also depend on specific brain regions [110,111].

### 4.2. The Role of Microglia in Alzheimer’s Disease

Alzheimer’s disease researchers postulate that the toxic Aβ plaques and phosphorylated tau play the leading roles in the AD pathogenesis [112,113], in which microglia activation-induced neuro-inflammation has emerged as an important factor [114]. At the initial stages of AD progression, microglia are beneficial and neuroprotective via the secretion of Aβ-degrading enzymes or by phagocytotic clearance of toxic Aβ plaques or extracellular tau [100]. In addition to phagocytosing amyloid-β plaques [100], accumulated evidence demonstrates that classically activated microglia are promoted during AD progression, leading to neuronal damage and secondary neurodegeneration [48]. In response to Aβ plaques, microglia play dual roles in the AD mechanism: (1) Aβ plaques activate the surveying microglia into neurotoxic microglia phenotype via the toll-like receptor and release a variety of pro-inflammatory molecules, including IL-6, IL-1β, ROS, and TNF-α, which induce astrocyte and neuronal damage with an increased level of apoptosis. (2) In turn, the activation of purinergic P2X7 receptors in microglia amplifies alternative activation. In other words, the dysfunction or over-activation of microglia in AD progression may lead to neuronal loss through Aβ accumulation from impaired amyloid clearance (Figure 6) [115,116]. Another study demonstrated that the pro-inflammatory phenotype of resident microglia impaired the phagocytosis of Aβ. Monocyte chemoattractant protein-1 (MCP-1) also activates or attracts microglia through the NF-kB/MAPK signaling pathway. One recent study reported that the low-density lipoprotein receptor-related protein 1 (LRP1) suppressed the microglial activation by modulating the activation of NF-kB and c-Jun N-terminal kinas (JNK) signaling pathways but not MAPK signaling [102]. Pro-inflammatory stimuli, such as LPS and oligomeric Aβ, down-regulated the LRP1 expression. However, NF-kB inhibitor restored the down-regulated LRP1 and eliminated the enhanced pro-inflammatory cytokine secretion stimulated by LPS. Taken together, targeting on dysfunctional microglial receptor, such as LRP1, which leads to microglial over-activation, has potential applications in discovering novel therapeutic strategies for treating AD patients. 

### 4.3. Generation of Microglia-Like Cells from Humna Pluripotent Stem Cells (hPSCs) 

Primary human microglia cells are not easily accessible. Recently, several in vitro studies focused on directing hPSC differentiation into microglia-like cells (Table 2) [47,117,118,119,120]. Differentiation microglia-like cells from hPSCs was firstly reported in 2016 [118]. Embryoid body (EB) formation in suspension using colony stimulating factor 1 (CSF1) and IL-34 in microglial differentiation medium was performed to obtain progenitors expressing VE-cadherin, c-kit, CD41, and CD235a, the markers of early yolk sac myelogenesis. Further differentiation generated CD11b+ IBA1+ semi-adherent cells with vacuolated and round morphology (about 8 weeks). The derived microglia-like progenitors expressed specific markers of microglia, including TMEM119, P2RY12, and IBA1, resembling those of fetal human microglia. The cells also expressed a panel of microglial-specific genes, including *PROS1, GAS6, MERTK, GPR34*, and *P2RY12*. The derived microglia-like progenitors also responded to IFN-γ and LPS by upregulating IL-6 and TNF-α at both protein and transcriptional levels, which supported their use as surrogates of human microglia.

Instead of EB formation, a robust protocol established by Douvaras et al. directed hiPSCs to differentiate into microglia-like cells (MG) in a myeloid inductive medium followed by treatments with IL-34 and granulocyte-macrophage colony-stimulating factor (GM-CSF). This method generated about 82% of CD14+ cells co-expressed with CX3CR1 after 25–50 days of monolayer culture [117]. The hiPSC-MGs expressed typical microglial markers, IBA1, CD11c (~95%), TMEM119, P2RY12 (~58%), CD11b (~94%), and CX3CR1 (~50%), and showed phagocytosis of microspheres (~90%) just like human primary microglia and macrophages. In addition, ADP-evoked intracellular Ca^2+^ transients were observed in hiPSC-MGs and primary microglia but not in macrophages, indicating the differences between microglia and peripheral macrophages. 

Another recent microglia differentiation protocol by Pandya et al. used astrocyte co-culture in the presence of serum. Human iPSC-MGs expressed HLA-DR, CD45, TREM2, and CX3CR1 in addition to CD11b and IBA1. In addition, MGs actively phagocytosed pHrodo red E. coli BioParticles (pHrodo) and produced ROS following stimulation with phorbol myristate acetate. Similarly, upregulated TNF-α secretion in the hiPSC-MG culture media was observed after LPS stimulation as compared to the media before LPS stimulation [47]. Taken together, the initial step of microglia differentiation is to induce hiPSC differentiation into the mesoderm instead of neuroectoderm even though microglia share the side-by-side residence in the human brain. Furthermore, the myeloid progenitors generated give rise to microglia progenitors under defined-factor medium or co-culturing with other cell types in the brain, such as astrocytes.

### 4.4. Neuronal-Microglial Crosstalk

Microglia interact with other cell types including neurons, astrocytes, and endothelial cells in the CNS. Recapitulation both the heterotypic cell–cell interactions and soluble factors in CNS is crucial for the investigation of the immune responses of microglia in vitro [49,126]. Co-culturing the hiPSC-derived macrophage-like cells with isogenic cortical neurons with IL-34 and GM-CSF recapitulated the microglia development *in vivo* and the co-cultured microglia showed the microglial-like transcriptome signature [121]. Microglia can sense neuronal activity through corresponding receptors responding to the signals exerted from neurons in the neuro-microglia environment and triggering the responses of cultured microglia [127]. For examples, in vitro studies demonstrated that the co-cultured microglia with neurons showed the enhanced motility with rapid ramified process and differential microglial gene expression [121,122]. Microglia-like cells were also observed to be integrated into the 3D organotypic neuroglial environment with dynamic motility [118]. Under LPS stimulation, co-cultured microglia migrate to form clusters and show the reduced ramification to adopt a more ameboid morphology. Similarly, under neuron injury, microglia-like cells surround the injury site and respond to the ATP and ADP released from the dying cells through P2RY12/13 (purinergic receptors) by migrating and encapsulating the damage area [118]. 

Microglia also express chemokine receptors. The important role of CD200-CD200R1 and CX3CL1-CX3CR1 chemokines signaling in neuronal–microglia interactions have also been demonstrated [122]. Damaged neuron-released CX3CL1 induce microglia migration toward endangered neurons [128]. In addition, the CX3CR1 in microglia showed the crucial role in the survival of layer V cortical neurons [129]. The exposure of microglial-like cells to CD200 and CX3CL1 modulate their response to Aβ oligomers by observing the increased expression of genes involved in phagocytosis of CNS substrates instead of AD-related genes, which indicates the inhibited microglia activities under neurodegenerative condition [122]. 

Given the brain region-dependent microglia diversity [110,111], it is believed that brain region-specific microenvironment promotes microglia function, and mutually microglia show selective regional sensitivity with neural cells. Our study co-cultured microglia-like cells with isogenic dorsal (D) or ventral (V) forebrain spheroids/organoids based on hiPSCs (Figure 7) (Unpublished data). Differential migration ability, intracellular Ca^2+^ signaling, and the response to pro-inflammatory stimuli (V-MG group had higher TNF-α and TREM2 expression, i.e., more pro-inflammatory) were observed. Transcriptome analysis exhibited 37 microglia-related genes that were differentially expressed in the MG and D-MG groups. In addition, the hybrid D-MG spheroids exhibited higher levels of immunoreceptor genes in activating members (e.g., *TREM1* and *CD300LB*), but the MG group contained higher levels of most genes in inhibitory members (e.g., *CD200R1, CD22, CD47*, and *SIRPA*).

Microglia can be derived innately along with cerebral organoids from mesodermal progenitors [125]. These organoid-grown microglia closely mimic the transcriptome and the immune response of the adult microglia. With the organoid development, a clear increase in expression of classical microglia markers was observed, i.e., AIF1/IBA1, CD68, ITGAM/CD11b, IRF8, TGFBR1, TGFBR2, TREM2, CX3CR1, HLADRA, C1QA, etc. Microglia were isolated from the organoids (called as oMGs) for RNAseq transcriptomic profiling. An increased expression of typical microglia genes AIF1, RUNX1, PTPRC, CX3CR1, TREM2, P2RY12, and TMEM119 was observed for day 119 vs. day 52 organoids. Secretion of IL-6, TNF-α, but not IL-10, was significantly increased upon LPS stimulation. 

Recent genetic evidence demonstrates that microglial genes that are crucial for microglia functions implicate a strong correlation with the risk of the late-onset AD [122,130]. The derived microglia cells from hiPSCs upregulated the expression of AD-related genes, including CD33, TREM2, APOE, and ABCA7, following Aβ oligomer exposure [122]. APOE is a significant factor functioning as a key mediator of microglia activation and Aβ deposition [131]. In vitro studies showed the regulator effects of CD33 and TREM2 on microglia phagocytosis of Aβ [97]. In addition, microglia expressed numerous genes associated with other neurological diseases, including Parkinson’s disease, amyotrophic lateral sclerosis, Huntington’s disease, and frontal temporal dementia [132,133,134].

## 5. Conclusions

3D brain organoids derived from hiPSCs provide an alternative to human brain models used for in vitro disease modeling. However, recapitulation of complex intercellular interactions is essential for the brain function. The establishment of neurovascular units in vitro promotes the functionality and maturation of cortical neurons. Reciprocally, direct neuronal-vascular contact promotes the capillary formation of co-cultured endothelial cells. Additionally, multicellular cultures with endothelial cells, neurons, astrocytes, and pericytes provide in vitro models of BBB using hiPSCs. Recapitulation of complex heterotypic cell-cell interactions in the BBB in vitro achieves characteristics comparable to physiological status. Overlapping with pericyte phenotype, hMSCs have shown benefits on neural tissue repair and angiogenesis mediated by paracrine signaling or cell-to-cell contacts. Furthermore, neuronal–microglia crosstalk plays the modulatory roles in regulating the immune response of microglia to different inflammatory stimuli in CNS. Over-activated microglia are involved in the progression of multiple neurodegenerative diseases. Therefore, carefully recapitulating the heterotypic cellular interactions in the human brain is beneficial for construction of a functional brain microenvironment with proper immune response. 

## Figures and Tables

**Figure 1 cells-08-00299-f001:**
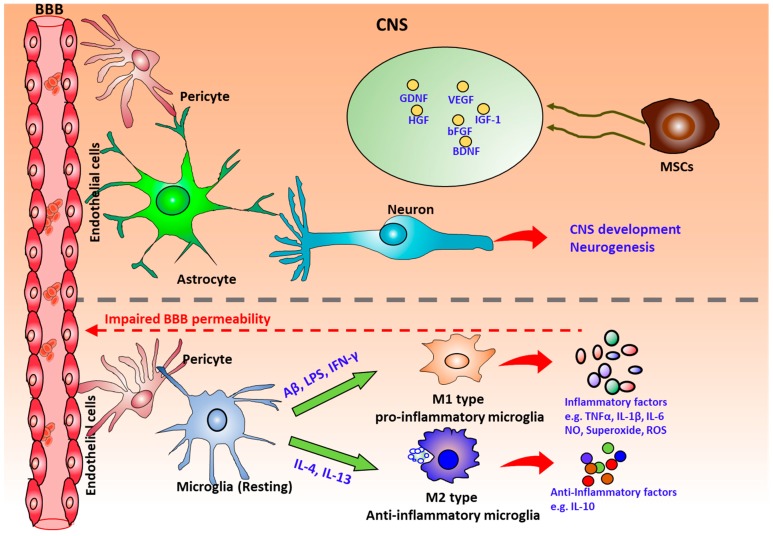
Cellular complexity in the central nervous system (CNS). The blood vessels in human brain form the blood–brain barrier (BBB) with endothelial cells, pericytes, astrocytes, and neurons. The endothelial cells also interact with microglia for immune response. Microglia have different activation pathways. Surveying microglia can be classically activated (M1) induced by lipopolysaccharide (LPS) or IFN-γ to release pro-inflammatory molecules, such as reactive oxygen species (ROS), TNF-α, IL-6, IL-1β, MMP and glutamate or alternatively activated (M2) by IL-4 or IL-13 to phagocytize pathogens and cell debris to induce an anti-inflammatory response with upregulation of IL-10 and arginase 1. Mesenchymal stem cells (MSCs), close to pericytes, secrete neurotrophic factors and angiogenesis factors.

**Figure 2 cells-08-00299-f002:**
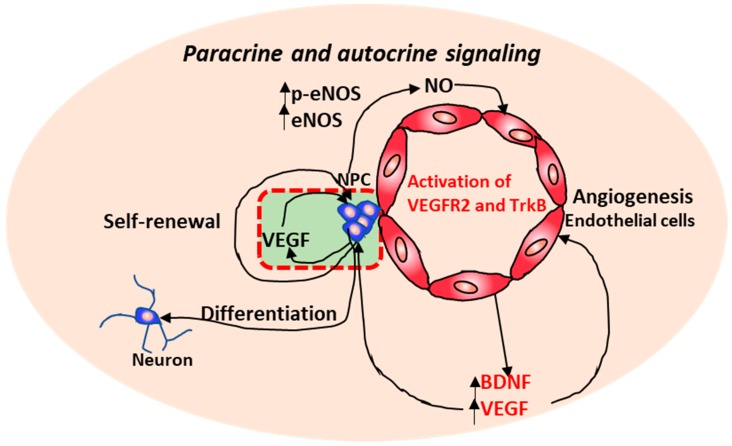
A working model of the dynamic interactions between neural progenitor cells (NPCs) and brain microvascular endothelial cells. VEGF: vascular endothelial growth factor; BDNF: brain-derived neurotrophic factor; NO: Nitric oxide; eNOS: endothelial nitric oxide synthase 3; VEGFR2: vascular endothelial growth factor receptor 2; TrkB: Tropomyosin receptor kinase B.

**Figure 3 cells-08-00299-f003:**
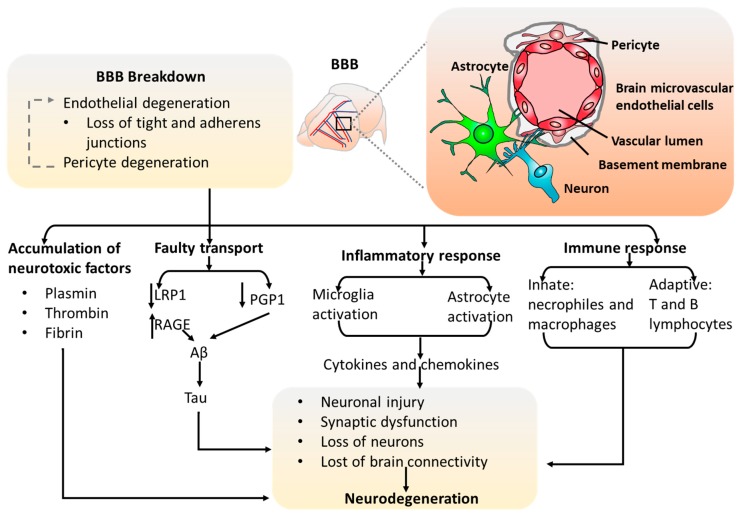
The blood–brain barrier (BBB) in vivo and the role in neural degeneration. The blood vessels in human brain form BBB with endothelial cells, pericytes, astrocytes, and neurons. BBB breakdown due to endothelial and pericyte degeneration leads to neural degeneration, associated with inflammatory response, loss of neurons, and synaptic dysfunction. LRP1: low density lipoprotein receptor-related protein 1; RAGE: receptor for advanced glycation end-products; PGP1: phosphatidylglycerolphosphate synthase 1.

**Figure 4 cells-08-00299-f004:**
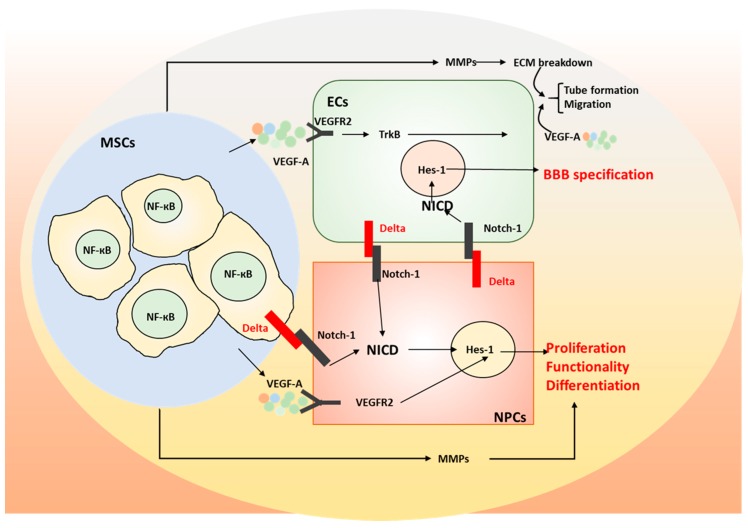
Schematic of the proposed signaling in spheroid human mesenchymal stem cells (MSCs). Interaction between VEGF and Notch signaling in neurovascular coupling. BBB: Blood-brain barrier; NPCs: neural progenitor cells; ECs: endothelial cells; MMP: matrix metalloproteinase; ECM: extracellular matrix; NICD: the intracellular domain of the notch protein; VEGF: vascular endothelial growth factor: VEGFR2: vascular endothelial growth factor receptor 2; TrkB: Tropomyosin receptor kinase B.

**Figure 5 cells-08-00299-f005:**
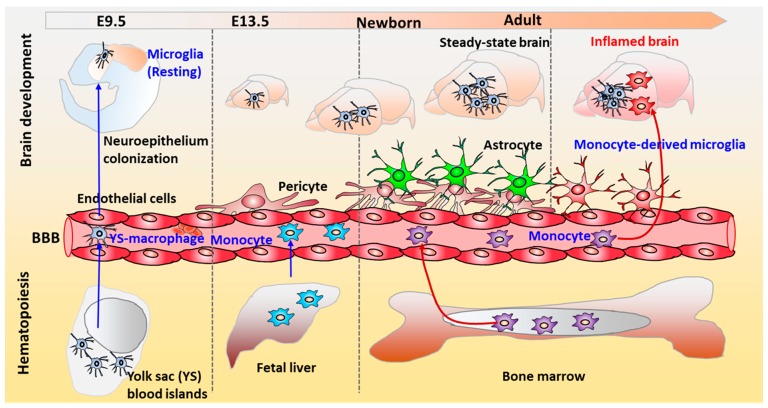
The physiological role of microglia in CNS development. Microglia originate from primitive hematopoietic stem cell at the extra-embryonic yolk sac (YS). Microglia migrate from the yolk sac into the central nervous system as the resident immune cells during brain development. Microglia regulate the neuronal activities and promote neuronal differentiation by secreting neurotrophic factors and anti-inflammatory molecules. In the developing brain, microglia can promote synaptic pruning and phagocytose neural progenitor cells.

**Figure 6 cells-08-00299-f006:**
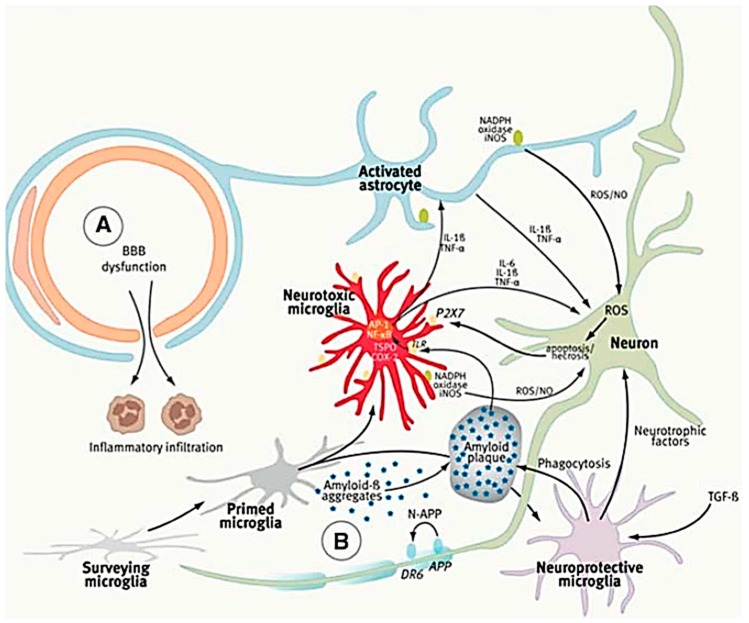
Vicious cycle of neuro-inflammation [116]. Aβ plaques activate the primed microglia into neurotoxic microglia phenotype via the toll-like receptor (TLR) and released a variety of pro-inflammatory molecules, including IL-6, IL-1β, and TNF-α, which induce astrocyte and neuronal damage with increased level of apoptosis. In turn, the activation of purinergic P2X7 receptors in microglia amplified alternative activation. Neuroprotective microglia are beneficial via the secretion of Aβ-degrading enzymes or by phagocytotic clearance of toxic Aβ plaques. From Jacobs et al, 2012.

**Figure 7 cells-08-00299-f007:**
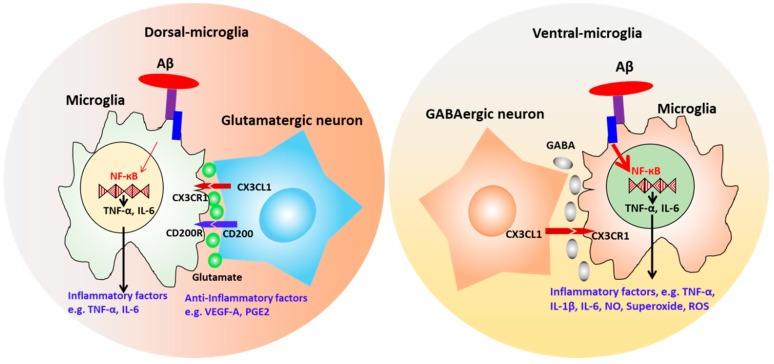
Neural-microglia interactions in hiPSC-based organoid models. Co-culturing the isogenic microglia with hiPSC-derived dorsal and ventral spheroids showed response to pro-inflammatory stimuli, Aβ42 oligomers. Dorsal-microglia group were less pro-inflammatory and showed higher anti-inflammatory cytokine secretion, while ventral-microglia group showed higher TNF-α expression under Aβ42 stimulation. All co-cultured spheroids stimulated cell proliferation and reduced reactive oxygen species (ROS) production, better resembling the tissue-specific microenvironment and the homeostasis.

**Table 1 cells-08-00299-t001:** Comparison of current in vitro BBB models using hiPSCs.

Models	Characterization	Culture System	Improvement	Ref.
Monoculture hiPSC-brain microvascular endothelial cells (BMECs)	Expression of tight junction proteins occludin and claudin-5 and p-glycoprotein and BBB glucose transporter GLUT-1; TEER was about 1450 Ωcm^2^ with astrocytes coculture.	2D neural and endothelial co-differentiation, providing a microenvironment resembling embryonic brain *in vitro.*	The first hPSC differentiation method that can reproducibly generate pure populations of EC with BBB properties.	Lippmann et al., 2012 [67]
Quadruple culture; hPSC-BMECs, pericytes, astrocytes, neurons	Expression of PECAM-1, GLUT-1, claudin-5 and occludin; TEER was about 5000 Ωcm^2^ with astrocytes coculture.	Adherent 2D transwell culture coated with collagen IV and fibronectin.	Retinoic acid (RA) enhanced BBB phenotypes in hPSC-BMECs.	Lippmann et al., 2014 [72]
Monoculture hiPSC-endothelial cells (ECs)	Expression of tight junction proteins, ZO-1, occludin, and claudin-5 and transporters proteins, PGG, LAT-1; Upregulated VECAD and TEER >2,000 Ωcm^2^ with the addition of retinoic acid.	Reproducible ECs induction protocol; Collagen IV and fibronectin coated surfaces.	The complex neurovascular environment should be employed.	Katt et al, 2016. [69]
Monoculture hiPSC-ECs	TEER >1,500 Ωcm^2^; Expression of BBB tight junction proteins ZO-1, Claudin-5, and Occludin, and BBB efflux transporters BCRP, MRP1, and PGP.	Adherent 2D transwell culture using derived BMECs.	Reproducibly consistent high TEER value. Evaluated cancer-targeting drug permeability.	Clark et al, 2016. [71]
Quadruple culture model hiPSC-ECs, hiPSC-NPCs, fetal brain astrocytes, pericytes	Robust BBB properties: TEER >2,500 Ωcm^2^; Upregulated BBB genes; ABCB1, SLC1A1, SLC2A1, and OCLN; Paracellular transport of small molecules were detected; *In vitro* recapitulation of transcellular passage of lipid-soluble agents.	Adherent 2D culture; Simultaneous co-culture effects;	Dynamic flow culture; Need to apply protocols for hiPSC-derived astrocytes and pericytes; BBB models from patient-specific hiPSCs.	Appelt-Menzel wt al, 2017. [38]
Triple culture hiPSC-ECs, hiPSC-derived neurons and astrocytes (1:3)	TEER ~ 886 Ωcm^2^; Slight increase in tight junction protein, occludin and cludin-5; Unchanged PGP efflux transporter activity.	EZ spheres; Isogenic BBB model.	A powerful tool for investigation of genetic disease modeling using patient-specific hiPSCs.	Canfield et al, 2017. [39]
Monoculture hiPSC-BMECs	Expressed GLUT-1, claudin-5, occludin, PECM-1 and VE-cadherin and consistently achieved TEER values exceeding 2500 Ωxcm^2^.	Adherent 2D transwell culture coated with collagen IV and fibronectin in E6 medium.	Reduced the differentiation time of iPSCs to BMECs from 13 to 8 days.	Hollmann et al., 2017 [73]
Monoculture hiPSC-BMECs	TEER > 3000 Ωcm^2^; BMECs phenotypes included tight junctions, low passive permeability, and polarized efflux transporters.	Adherent 2D transwell culture coated with collagen IV and fibronectin.	A facile, chemically defined method to differentiate hPSCs to BMECs via sequential Wnt and RA activation.	Qian et al., 2017 [74]
Monoculture HD hiPSC-BMECs	HD-BMECs had intrinsic impairments in angiogenic potential and drug efflux, and showed impaired paracellular and transcellular barrier properties.	Adherent 2D transwell culture coated with collagen IV and fibronectin.	Reduce disease burden and assess BBB penetration of drugs for HD.	Lim et al., 2017 [75]
Coculture hiPSC-BMECs, rat astrocytes	TEER levels peaked above 4000 Ωcm^2^ and were sustained above 2000 Ωcm^2^ up to 10 days.	Microfluidic platform.	A microfluidic BBB model mimicked *in vivo* BBB integrity and compound permeability.	Wang et al., 2017 [76]
Coculture MCT8 hiPSC-BMECs, hiPSC-NPCs	MCT8-deficient BMECs showed defects in thyroid hormone transport.	BMECs were cocultured with EZ sphere-derived neural cells in transwell.	A platform to test candidate drug transport across the diseased BBB.	Vatine et al., 2017 [77]
Triculture hiPSC-ECs, pericytes, astrocytes	The BBB model exhibited perfusable and selective microvasculature, permeability lower than conventional *in vitro* models, and similar to *in vivo* measurements in rat brain.	PDMS microfluidic system in fibrin gel.	A robust and physiologically relevant BBB microvascular model.	Campisi eti al., 2018 [65]
Coculture hiPSC-BMECs, NPC-astrocytes	TNF-α and IL-6 treatment impaired BBB integrity; Coculture with NPC-astrocytes improved TEER.	Transwell culture system.	The model mimicked cellular responses to inflammation at the BBB.	Mantle et al., 2018 [78]
Four cell types: hiPSC-BMECs, hiPSC-astrocytes, neurons, and pericytes	BMECs in coculture model showed high TEER and functional efflux; Whole genome expression profiling revealed upregulation of tight junction proteins.	Transwell culture coated with collagen IV and fibronectin.	Whole genome analysis about hiPSC-BBB model.	Delsing et al., 2018 [79]
Coculture hiPSC-ECs, hiPSC-NPCs	significant barrier integrity with tight junction protein expression, an effective permeability to sodium fluorescein and higher TEER value.	3D printed electrospun PLGA scaffold.	BBB model reduced the penetration of Aβ oligomer into the neurons from hiPSC-NPCs.	Qi et al., 2018 [66]
Six cell types: hiPSC-microglia, oligodendrocyte, neurons, human primary BMECs, astrocytes, pericytes	Spheroids showed expression of tight junctions, adherens junctions, adherens junction-associated proteins and cell specific markers.	3D cortex spheroid.	Organoid model formed a functional BBB.	Nzou et al., 2018 [20]
Coculture AD hiPSC-BMECs or healthy hiPSC-BMECs	Expression of tight junction proteins occludin and claudin-5 and p-glycoprotein and BBB glucose transporter GLUT-1. Adm BMECs showed no difference in TEER value and permeability compared to control.	Collagen type I microvessels in PDMS microfluidic chip.	Physiological BBB microvessel model to study barrier function.	Linville et al., 2019 [80]

Notes: hiPSC, human-induced pluripotent stem cells; BBB, Blood-brain barrier; BMECs, brain microvascular endothelial cells; EC, endothelial cells; NPCs, neural progenitor cells; HD, Huntington’s disease; MCT8, monocarboxylate transporter 8; PLGA, poly lactic-co-glycolic acid; PDMS, polydimethylsiloxane; TEER, transendothelial electrical resistance.

**Table 2 cells-08-00299-t002:** Comparison of current protocols for microglia differentiation from hPSCs.

Cell Source	Culture System	Yield (MG/PSC)	Phenotypic and Functional Characterization	Ref.
hiPSCs	Monolayer serum-free culture using IL-34/GM-CSF	2.24	iPSC-MGs expressed typical microglial markers, IBA1, CD11c (~95%), TMEM119, P2RY12 (~58%), CD11b (~94%) and CX3CR1 (~50%); iPSC-MGs showed phagocytosis of microspheres (~90%) as human primary microglia and macrophages; ADP-evoked intracellular Ca^2+^ transients were observed in iPSC-MGs and primary microglia but not in macrophages.	Douvaras et al, 2017 [117]
hESCs or hiPSCs	EBs using serum-free culture	0.5–4.0	Expressed specific markers of microglia, including TMEM119, P2RY12, and IBA1; responded to IFN-γ and LPS by upregulating IL-6, TNF-α at both protein and transcriptional levels.	Muffat et al, 2016 [118]
hiPSCs	Co-culture with astrocytes on monolayer	2–3	Human iPS-MGs expressed HLA-DR, CD45, TREM-2 and CX3CR1 in addition to CD11b and IBA1; MGs phagocytosed pHrodo red E. coli BioParticles (pHrodo) and produced reactive oxygen species (ROS) following stimulation with phorbol myristate acetate.	Pandya et al, 2017 [47]
hiPSCs H9 hESCs	FACS-sorted CD43+ cells, with M-CSF, IL-34, TGF-β1	125 CD43+ cells/PSC	Similar transcriptome and identical phagocytosis ability compared to iPSC-MG of previous protocols. “iPSC-microglia 2.0” engrafted well into xenotransplantation compatible MITRG mice.	McQuade wt al, 2018 [119]
hiPSCs	Co-culture with hiPSC-cortical neurons and IL-34 and GM-CSF	40	Expressed key surface protein markers; Positive for P2RY12, GPR34, <ERTK, C1QA, PROS1, GAS6, TMEM119 and TREM2; Phagocytic and release microglia-relevant cytokines and upregulate homeostatic function pathways.	Haenseler et al., 2017 [121]
hiPSCs	FACS-sorted CD43+ cells, with MCSF, IL-34, TGFβ1, insulin, CD200 and CX3CL1	30–40	Positive for MERTK, ITGB5, CX3CR1, TGFβR1, PROS1, P2RY12, PU.1 and TREM2; Transcriptome comparable to adult and fetal human microglia; Secreted cytokines, respond to inflammatory stimuli; calcium transients, phagocytosisfor Aβ fibrils and tau oligomers; transplanted into transgenic mice and human brain organoids, resembled microglia *in vivo*.	Abud et al., 2017 [122]
hiPSCs	EBs in hypoxia with BMP4, activin A, FGF2, VEGF, CSF-1, and IL-3	unknown	Positive for IBA1 and CX3CR1; Phagocytosis of beads and Aβ; FACS-sorted CD45+ CD11b+, coculture with hiPSC-neurons.	Takata et al., 2017 [123]
hPSC-macrophage precursors	EB, using GM-CSF and IL-34	30–40	Positive for IBA1, CD45, TREM2; Whole-transcriptome showed similar signature to primary microglia; Mutant TREM2 caused to immature form of microglia without typical proteolysis.	Brownjohn et al., 2018 [124]
hiPSCs	Mesodermal progenitors developed into microglia-like cells within cerebral organoids	unknown	Positive for PU.1, CSF1R, CD68, IBA1, IRF8, TREM2, CXCR1, C1QA; Transcriptome analysis showed similar signature to primary microglia; Mediated phagocytosis and synaptic activities.	Ormel et al., 2018 [125]

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
