# Peer review of "Studying Heterotypic Cell–Cell Interactions in the Human Brain Using Pluripotent Stem Cell Models for Neurodegeneration"

_cells, 2019, doi:10.3390/cells8040299_

Round 1

Reviewer 1 Report

In this manuscript, authors well reviewed about the modeling neurodegenneraive diseases by taking advantage of various cell-cell interactions that are derived from human pluripotent stem cells(iPSC). Especially, they comprehensively described about roles of microglia, BBB and etc.

There are some points to be addressed before the manuscript gets acceptable as described below.

1.    Microglia and macrophages have been classified as pro-inflammatory (M1) or anti-inflammatory (M2) (Martinez and Gordon, 2014; Michell-Robinson et al., 2015; Walker and Lue, 2015). However, this strict classification is considered to be no longer valid (Ransohoff, 2016). In fact, the terms of M1 and M2 are used in this manuscript. On the other hand, as a recent trend, Butovsky et al. (2018) propose to use the terms “homeostatic miscroglia” and “disease-associated microglia” to provide insight into how these cells are regulated in health and disease and how they contribute to the maintenance of the neural environment.  This issue should be discussed in the text.

2.    As regards the dorsal and ventral microglia described in Figure 7. (Neural-microglia interactions using hiPSC-based organoid models), they seem to be very artificial definition. Please provide more insights how they are relevant to the in vivo situation.

Author Response

Reviewer #1

In this manuscript, authors well reviewed about the modeling neurodegenneraive diseases by taking advantage of various cell-cell interactions that are derived from human pluripotent stem cells(iPSC). Especially, they comprehensively described about roles of microglia, BBB and etc.

There are some points to be addressed before the manuscript gets acceptable as described below.

1.     Microglia and macrophages have been classified as pro-inflammatory (M1) or anti-inflammatory (M2) (Martinez and Gordon, 2014; Michell-Robinson et al., 2015; Walker and Lue, 2015). However, this strict classification is considered to be no longer valid (Ransohoff, 2016). In fact, the terms of M1 and M2 are used in this manuscript. On the other hand, as a recent trend, Butovsky et al. (2018) propose to use the terms “homeostatic miscroglia” and “disease-associated microglia” to provide insight into how these cells are regulated in health and disease and how they contribute to the maintenance of the neural environment.  This issue should be discussed in the text.

Response:  As the reviewer suggested, we include the recent literature about the classification of microglia polarization.  We added a paragraph (and the related references) on Page 13 about the classification of microglia immunophenotype.

References

Butovsky, O.; Weiner, H. L., Microglial signatures and their role in health and disease. Nat Rev Neurosci 2018, 19, 622-635

Ransohoff, R. M., How neuroinflammation contributes to neurodegeneration. Science 2016, 353, 777-783.

Walker, D. G.; Lue, L. F., Immune phenotypes of microglia in human neurodegenerative disease: challenges to detecting microglial polarization in human brains. Alzheimers Res Ther 2015, 7, 56.

2.    As regards the dorsal and ventral microglia described in Figure 7. (Neural-microglia interactions using hiPSC-based organoid models), they seem to be very artificial definition. Please provide more insights how they are relevant to the in vivo situation.

Response:  Based on the point 1 above, we revised the description in Figure 7 legends.  The dorsal group is less pro-inflammatory than the ventral group, and the illustration is based on our unpublished data.  For in vivo situation, for example, Alzheimer’s disease, the ventral cortex part (and hippocampus) may be more disease-associated.

Reference:

Grabert, K.; Michoel, T.; Karavolos, M. H.; Clohisey, S.; Baillie, J. K.; Stevens, M. P.; Freeman, T. C.; Summers, K. M.; McColl, B. W., Microglial brain region-dependent diversity and selective regional sensitivities to aging. Nat Neurosci 2016, 19, 504-516.

Kanekiyo, T.; Zhang, J.; Liu, Q.; Liu, C. C.; Zhang, L.; Bu, G., Heparan sulphate proteoglycan and the low-density lipoprotein receptor-related protein 1 constitute major pathways for neuronal amyloid-beta uptake. The Journal of neuroscience: the official journal of the Society for Neuroscience 2011, 31, 1644-1651.

Reviewer 2 Report

Manuscript title: Studying Heterotypic Cell-Cell Interactions in Human Brain Using Pluripotent Stem Cell Models for Neurodegeneration.

In their manuscript, Song et al. point out the importance of investigating the contribution of non-neuronal cells in neurodegeneration presenting several examples of hIP cells-derived models. They also enlighten the need to introduce non-neuronal cells in the currently available 3-dimensional (3D) cellular models to better investigate biology of the brain in physiological and pathological conditions. Specifically, they focus on mesenchymal stem cells, endothelial cells and microglia. This review is tackling a very hot topic in neuroscience especially in the field of induced pluripotent stem cells models and it is offering a tabular overview of the iPS cell-derived in vitro models of blood brain barrier and of the most recent protocols to reprogram iPS cells into microglia. Nevertheless, throughout the manuscript several sentences are not clear, and the choice of many references was not optimal, thus making the whole review not fluent and -in some parts- hard to comprehend.

Here some examples:

Major changes:

- As general remark, the introduction is fragmented, and it is difficult to link the different paragraphs in a logic outline. The authors should pay more attention to make a clear distinction between cerebral organoids and 3D cultures of other organs.

- The authors make a valid point on the necessity to integrate non-neuronal cells in the cerebral 3D cultures. Nevertheless, the discussion of this issue (lines 43-58) is not clear. Specifically, they should at least describe the pro- and cons- of the experimental approach of Takebe et al. and if it could improve the current neuronal 3D culture models.

- At lines 88-89, the sentence should be rephrased. Despite microglia is of pivotal importance in the processes of brain development and homeostasis the implementation of this cellular population within cerebral organoids cannot provide -alone- a full recapitulation of the “functions of the human brain”.

- At line 90, it is not clear what the authors want to state. The motivation to investigate the heterotypic interactions between cells in the brain is deriving by the necessity to improve the complexity of the current in vitro systems and obtain models with a higher translational impact. The inefficient neural differentiation of hiPSCs do not represent the issue, especially if we consider that the cells mentioned in this review derive from a mesodermal and not neuroectodermal lineage.

- Infiltrating peripheral macrophages during pathologic conditions can reach the brain and increase the pool of active phagocytic cells in the brain, together with microglia. However, they maintain their own identity and do not simply become microglia (reviewed in Prinz et al 2018 - Ontogeny and homeostasis of CNS myeloid cells). Thus, authors should rephrase the sentence at line 358 as follow:

“...which is increasing the pool of macrophages within the brain.”

- At line 228, the sentence is not clear.

- At line 363, CX3CR1 was probably misspelled, since this is the fractalkine (CX3CL1) receptor. Probably the authors wanted to mention the activity of CXCR3, the receptor of CXCL10.

- In Figure 5 caption, authors should mention also synaptic pruning and phagocytosis of NPCs as microglia functions in the developing brain.

- At lines 382-388, the sentence is not clear, especially regarding the “dual roles in AD metabolism”.

- The literature cited about cortical/cerebral organoids is scarce. The authors should present other recent original works in line 38 and mention the use of 3D cerebral organoids in the study of psychiatric disorders (lines 41-41).

- The references 40, 41 and 42 are unpublished findings from the author’s laboratory and should not be presented in the text as published and peer-reviewed results until the publication. I highly recommend to present them as “Unpublished data”.

- At line 52, the authors should check the references used. While ref. 7 is a general review of 2014 on the topic of organoids, the work of Pasca et al. and Birey et al. do not present cerebral organoids co-cultured with endothelial nor microglial cells. The authors should separately cite original papers making a distinction between cortical and cerebral organoids. In addition, the work of Schwartz and collaborators report the use 3D neural constructs on biomaterials rather than 3D organoids. Lastly, the paper mentioned at the end of the sentence (ref. 10) deals with the use of iPS cell derived organ bud for liver transplant. All in all, these references are misleading and are not integrating within the paragraph.

- At lines 59-60, a proper reference is missing. In the paragraph it is not clear if the context is referred to 3D cultures in general or to cerebral organoids.

- At line 86, a reference is missing.

- The authors should check reference 51, because it seems not to fit with the text.

- At line 151, the authors should reconsider the references cited. Ref. 12 do not present co-culture of NPC and endothelial cells in a 3D context; ref. 19 is a review mainly focused on MSC and cancer; ref. 20 is a systematic review on differentiation of stem cells into pericytes.

- At line 320, authors should check if the citation of ref. 13 is correct, since in Schwartz et al. 3D spheroids of hMSC are not used.

- The reference on microglia at line 348 (Ferrer, I.; Bernet, E.; Soriano, E.; Del Rio, T.; Fonseca, M., Naturally occurring cell death in the cerebral 761 cortex of the rat and removal of dead cells by transitory phagocytes. Neurosci 1990, 39, 451-458.)  is outdated. More recent overviews of microglia physiology and function should be used.

Minor changes:

- At line 194, please replace “In addition” with “Moreover” to avoid a repetition in the sentence.

- At line 228, please correct “calcium-bind protein” to “calcium-binding protein”.

- Figure 7 legend is not formatted as the overall manuscript.

- In the tables not all the sentences finish with a dot. The formatting should be uniformed.

Author Response

Reviewer #2

In their manuscript, Song et al. point out the importance of investigating the contribution of non-neuronal cells in neurodegeneration presenting several examples of hIP cells-derived models. They also enlighten the need to introduce non-neuronal cells in the currently available 3-dimensional (3D) cellular models to better investigate biology of the brain in physiological and pathological conditions. Specifically, they focus on mesenchymal stem cells, endothelial cells and microglia. This review is tackling a very hot topic in neuroscience especially in the field of induced pluripotent stem cells models and it is offering a tabular overview of the iPS cell-derived in vitro models of blood brain barrier and of the most recent protocols to reprogram iPS cells into microglia. Nevertheless, throughout the manuscript several sentences are not clear, and the choice of many references was not optimal, thus making the whole review not fluent and -in some parts- hard to comprehend.

Response:  We would like to thank the reviewer for the careful reading in order to improve our manuscript. We apologize for the improper references and the unclear sentences.  We made the careful corrections and clarified our points.  Therefore, we think that the revision made the necessary improvement.

Here some examples:

Major changes:

- As general remark, the introduction is fragmented, and it is difficult to link the different paragraphs in a logic outline. The authors should pay more attention to make a clear distinction between cerebral organoids and 3D cultures of other organs.

Response:  We agree with the reviewer that link the different paragraphs in a logic outline is important.  As the reviewer suggested (please see the responses below), we revised the introduction and the references to reduce the confusion about cerebral organoids and 3D cultures of other organs.

- The authors make a valid point on the necessity to integrate non-neuronal cells in the cerebral 3D cultures. Nevertheless, the discussion of this issue (lines 43-58) is not clear. Specifically, they should at least describe the pro- and cons- of the experimental approach of Takebe et al. and if it could improve the current neuronal 3D culture models.

Response:  This paragraph is to illustrate the importance of heterotypic cell-cell interactions.  As the reviewer suggested, we revised these sentences to describe the cons- of the experimental approach of Takebe et al: “This approach simply mixed the cells of different types and the cell organization and tissue structure are uncontrolled.  In addition, the endothelial cells are not organ-specific.”

- At lines 88-89, the sentence should be rephrased. Despite microglia is of pivotal importance in the processes of brain development and homeostasis the implementation of this cellular population within cerebral organoids cannot provide -alone- a full recapitulation of the “functions of the human brain”.

Response: As the reviewer suggested, we revised this sentence, which now reads “However, most of recent brain organoids from hiPSCs lack several critical components (e.g., vascular cells, BBB feature) in the human brain, one of which is the microglia [1], so fully mimicking the function of human brain has not been achieved.”

- At line 90, it is not clear what the authors want to state. The motivation to investigate the heterotypic interactions between cells in the brain is deriving by the necessity to improve the complexity of the current in vitro systems and obtain models with a higher translational impact. The inefficient neural differentiation of hiPSCs do not represent the issue, especially if we consider that the cells mentioned in this review derive from a mesodermal and not neuroectodermal lineage.

Response: As the reviewer suggested, we revised this sentence, which now reads “Inefficient neural function of cells or culture models derived from hiPSCs motivates recent investigations of complex heterotypic interactions of different cell types to recapitulate cellular behaviors in vivo [1, 49].”

- Infiltrating peripheral macrophages during pathologic conditions can reach the brain and increase the pool of active phagocytic cells in the brain, together with microglia. However, they maintain their own identity and do not simply become microglia (reviewed in Prinz et al 2018 - Ontogeny and homeostasis of CNS myeloid cells). Thus, authors should rephrase the sentence at line 358 as follow: “...which is increasing the pool of macrophages within the brain.”

Response:  As the reviewer suggested, we revised this sentence, which now reads “In the meantime, the bone marrow-derived myeloid cells may penetrate the compromised blood-brain barrier and differentiate into microglia to infiltrate the CNS, which might be the second origin of microglia [103].  However, it was also suggested that infiltrating peripheral macrophages during pathologic conditions can reach the brain and increase the pool of active phagocytic cells in the brain, together with microglia, but they maintain their own identity and do not simply become microglia [104].”

We also included the referred reference.

- At line 228, the sentence is not clear.

Response: As the reviewer suggested, we revised this sentence, which now reads “Pericytes, directly contact with BMECs, are a heterogeneous MSC-like population in terms of morphology and marker expression, and contribute to BBB permeability and neural inflammation [81].”

Reference:

Rustenhoven, J.; Jansson, D.; Smyth, L. C.; Dragunow, M., Brain Pericytes As Mediators of Neuroinflammation. Trends Pharmacol Sci 2017, 38, 291-304.

- At line 363, CX3CR1 was probably misspelled, since this is the fractalkine (CX3CL1) receptor. Probably the authors wanted to mention the activity of CXCR3, the receptor of CXCL10.

Response:  As the reviewer suggested, we corrected this error.  It should be CXCR3, the receptor of CXCL10.

- In Figure 5 caption, authors should mention also synaptic pruning and phagocytosis of NPCs as microglia functions in the developing brain.

Response:  As the reviewer suggested, we revised Figure 5 caption to include the microglia function about synaptic pruning and phagocytosis of NPCs in the developing brain.

- At lines 382-388, the sentence is not clear, especially regarding the “dual roles in AD metabolism”.

Response:  As the reviewer suggested, we revised this sentence, which now reads “accumulated evidences demonstrate that classically activated microglia are promoted during AD progression, leading to neuronal damage and secondary neurodegeneration [47]. In response to Aβ plaques, microglia play dual roles in AD mechanism: (1) Aβ plaques activate the surveying microglia into neurotoxic microglia phenotype via the toll-like receptor and release a variety of pro-inflammatory molecules, including IL-6, IL-1β, ROS, and TNF-α, which induce astrocyte and neuronal damage with an increased level of apoptosis. (2) In turn, …..”

- The literature cited about cortical/cerebral organoids is scarce. The authors should present other recent original works in line 38 and mention the use of 3D cerebral organoids in the study of psychiatric disorders (lines 41-41).

Response:  As the reviewer suggested, we included more literature about cortical/cerebral organoids, and those used for the study of psychiatric disorders.

1. Klaus, J.; Kanton, S.; Kyrousi, C.; Ayo-Martin, A. C.; Di Giaimo, R.; Riesenberg, S.; O'Neill, A. C.; Camp, J. G.; Tocco, C.; Santel, M.; Rusha, E.; Drukker, M.; Schroeder, M.; Gotz, M.; Robertson, S. P.; Treutlein, B.; Cappello, S., Altered neuronal migratory trajectories in human cerebral organoids derived from individuals with neuronal heterotopia. Nat Med 2019.

2. Stachowiak, E. K.; Benson, C. A.; Narla, S. T.; Dimitri, A.; Chuye, L. E. B.; Dhiman, S.; Harikrishnan, K.; Elahi, S.; Freedman, D.; Brennand, K. J.; Sarder, P.; Stachowiak, M. K., Cerebral organoids reveal early cortical maldevelopment in schizophrenia-computational anatomy and genomics, role of FGFR1. Transl Psychiatry 2017, 7, 6.

3. Gonzalez, C.; Armijo, E.; Bravo-Alegria, J.; Becerra-Calixto, A.; Mays, C. E.; Soto, C., Modeling amyloid beta and tau pathology in human cerebral organoids. Mol Psychiatry 2018, 23, 2363-2374.

4. Ilieva, M.; Fex Svenningsen, A.; Thorsen, M.; Michel, T. M., Psychiatry in a Dish: Stem Cells and Brain Organoids Modeling Autism Spectrum Disorders. Biol Psychiatry 2018, 83, 558-568.

5. Bagley, J. A.; Reumann, D.; Bian, S.; Levi-Strauss, J.; Knoblich, J. A., Fused cerebral organoids model interactions between brain regions. Nat Methods 2017, 14, 743-751.

6. Pasca, A. M.; Sloan, S. A.; Clarke, L. E.; Tian, Y.; Makinson, C. D.; Huber, N.; Kim, C. H.; Park, J. Y.; O'Rourke, N. A.; Nguyen, K. D.; Smith, S. J.; Huguenard, J. R.; Geschwind, D. H.; Barres, B. A.; Pasca, S. P., Functional cortical neurons and astrocytes from human pluripotent stem cells in 3D culture. Nature methods 2015, 12, 671-678.

7. Yoon, S. J.; Elahi, L. S.; Pasca, A. M.; Marton, R. M.; Gordon, A.; Revah, O.; Miura, Y.; Walczak, E. M.; Holdgate, G. M.; Fan, H. C.; Huguenard, J. R.; Geschwind, D. H.; Pasca, S. P., Reliability of human cortical organoid generation. Nat Methods 2019, 16, 75-78.

- The references 40, 41 and 42 are unpublished findings from the author’s laboratory and should not be presented in the text as published and peer-reviewed results until the publication. I highly recommend to present them as “Unpublished data”.

Response:  As the reviewer suggested, we removed references 40, 41, and 42, and referred them as “Unpublished data”.

- At line 52, the authors should check the references used. While ref. 7 is a general review of 2014 on the topic of organoids, the work of Pasca et al. and Birey et al. do not present cerebral organoids co-cultured with endothelial nor microglial cells. The authors should separately cite original papers making a distinction between cortical and cerebral organoids. In addition, the work of Schwartz and collaborators report the use 3D neural constructs on biomaterials rather than 3D organoids. Lastly, the paper mentioned at the end of the sentence (ref. 10) deals with the use of iPS cell derived organ bud for liver transplant. All in all, these references are misleading and are not integrating within the paragraph.

Response: As the reviewer suggested, we revised the related references.  We replaced the mentioned references with those related to cortical or cerebral organoids, and those with multiple cell types.

- At lines 59-60, a proper reference is missing. In the paragraph it is not clear if the context is referred to 3D cultures in general or to cerebral organoids.

Response: As the reviewer suggested, we added two references.

Teixeira, F. G.; Carvalho, M. M.; Sousa, N.; Salgado, A. J., Mesenchymal stem cells secretome: a new paradigm for central nervous system regeneration? Cellular and molecular life sciences : CMLS 2013, 70, 3871-3882.

Teixeira, F. G.; Carvalho, M. M.; Neves-Carvalho, A.; Panchalingam, K. M.; Behie, L. A.; Pinto, L.; Sousa, N.; Salgado, A. J., Secretome of mesenchymal progenitors from the umbilical cord acts as modulator of neural/glial proliferation and differentiation. Stem cell reviews 2015, 11, 288-297.

This paragraph is more for 3D cultures in general.  So we revised the first sentence of this paragraph.

- At line 86, a reference is missing.

Response:  As the reviewer suggested, we added two references.

Lloyd, A. F.; Davies, C. L.; Miron, V. E., Microglia: origins, homeostasis, and roles in myelin repair. Curr Opin Neurobiol 2017, 47, 113-120.

Cunningham, C. L.; Martinez-Cerdeno, V.; Noctor, S. C., Microglia regulate the number of neural precursor cells in the developing cerebral cortex. J Neurosci 2013, 33, 4216-4233.

- The authors should check reference 51, because it seems not to fit with the text.

Response: As the reviewer suggested, reference 51 was replaced with a more relevant reference.

Li, Q.; Ford, M. C.; Lavik, E. B.; Madri, J. A., Modeling the neurovascular niche: VEGF- and BDNF-mediated cross-talk between neural stem cells and endothelial cells: an in vitro study. J Neurosci Res 2006, 84, 1656-1668.

- At line 151, the authors should reconsider the references cited. Ref. 12 do not present co-culture of NPC and endothelial cells in a 3D context; ref. 19 is a review mainly focused on MSC and cancer; ref. 20 is a systematic review on differentiation of stem cells into pericytes.

Response: As the reviewer suggested, we removed references 12, 19, and 20, and added the reference 2.

Mansour, A. A.; Goncalves, J. T.; Bloyd, C. W.; Li, H.; Fernandes, S.; Quang, D.; Johnston, S.; Parylak, S. L.; Jin, X.; Gage, F. H., An in vivo model of functional and vascularized human brain organoids. Nat Biotechnol 2018, 36, 432-441.

- At line 320, authors should check if the citation of ref. 13 is correct, since in Schwartz et al. 3D spheroids of hMSC are not used.

Response: As the reviewer suggested, reference 13 is replaced with the following reference.

Sart, S.; Tsai, A.-C.; Li, Y.; Ma, T., Three-dimensional aggregates of mesenchymal stem cells: cellular mechanisms, biological properties, and applications. Tissue Engineering Part B Reviews 2014, 20, 365-380.

- The reference on microglia at line 348 (Ferrer, I.; Bernet, E.; Soriano, E.; Del Rio, T.; Fonseca, M., Naturally occurring cell death in the cerebral 761 cortex of the rat and removal of dead cells by transitory phagocytes. Neurosci 1990, 39, 451-458.)  is outdated. More recent overviews of microglia physiology and function should be used.

Response: As the reviewer suggested, it is replaced with more recent reference as seen in the following.

Wolf, S. A.; Boddeke, H. W.; Kettenmann, H., Microglia in Physiology and Disease. Annu Rev Physiol 2017, 79, 619-643.

Minor changes:

- At line 194, please replace “In addition” with “Moreover” to avoid a repetition in the sentence.

Response: As the reviewer suggested, it is corrected.

- At line 228, please correct “calcium-bind protein” to “calcium-binding protein”.

Response: As the reviewer suggested, it is corrected.

- Figure 7 legend is not formatted as the overall manuscript.

Response: As the reviewer suggested, it is formatted.

- In the tables not all the sentences finish with a dot. The formatting should be uniformed.

Response:  As the reviewer suggested, the formatting was revised to be uniformed for the Tables.

Round 2

Reviewer 2 Report

The authors have improved their manuscript by replying on a point-to-point base to the issues raised in the first review of their work. The references have been improved and the sentences clarified. However, there are still some issues that should be corrected as listed below:

- At lines 61-62 the authors should be more cautious, since references 23 and 24 do not present examples of the modulation of MSC secretome on 3D cultures. They should cite more appropriate works or mitigate the sentence as following: “…MSC secretome might modulate the neurogenic niche and promote neural differentiation through trophic effects.”

- At lines 361-363 the reference 103 (Yang, L.; Liu, C.-C.; Zheng, H.; Kanekiyo, T.; Atagi, Y.; Jia, L.; Wang, D.; N’songo, A.; Can, D.; Xu, H., LRP1 modulates the microglial immune response via regulation of JNK and NF-κB signaling pathways. J Neuroinflammation 2016, 13, 304.) do not present any example of infiltration and differentiation of bone marrow-derived myeloid cells in the CNS. The authors should find a proper reference supporting their statement or delete the sentence.

- At lines 453-455 I would suggest to remove the sentence "even though microglia share the side-by-side residence in the human brain". As correctly stated by the authors in paragraph 4.1, microglia generate from the yolk sac and invade the brain during neurodevelopment. The necessity to use differentiation protocols toward the mesodermal lineage is simply due to the different embryonic origin of microglia and neuronal cells. Sharing the residence in the human brain with neurons cannot be linked to their distinct developmental origin. 

Minor comments:

- at line 54 delete “recently” to avoid a repetition.

- at lines 94 delete "cells or" to avoid redundance in the sentence.

- In the introduction some sentences are still difficult to read and are not properly written (i.e. - at lines 91-93). I would recommend to have the manuscript checked by a native English speaker.

Author Response

The authors have improved their manuscript by replying on a point-to-point base to the issues raised in the first review of their work. The references have been improved and the sentences clarified. However, there are still some issues that should be corrected as listed below:

- At lines 61-62 the authors should be more cautious, since references 23 and 24 do not present examples of the modulation of MSC secretome on 3D cultures. They should cite more appropriate works or mitigate the sentence as following: “…MSC secretome might modulate the neurogenic niche and promote neural differentiation through trophic effects.”

Response:  As the reviewer suggested, we revised the sentence.

- At lines 361-363 the reference 103 (Yang, L.; Liu, C.-C.; Zheng, H.; Kanekiyo, T.; Atagi, Y.; Jia, L.; Wang, D.; N’songo, A.; Can, D.; Xu, H., LRP1 modulates the microglial immune response via regulation of JNK and NF-κB signaling pathways. J Neuroinflammation 2016, 13, 304.) do not present any example of infiltration and differentiation of bone marrow-derived myeloid cells in the CNS. The authors should find a proper reference supporting their statement or delete the sentence.

Response:  As the reviewer suggested, we removed the sentence.

- At lines 453-455 I would suggest to remove the sentence "even though microglia share the side-by-side residence in the human brain". As correctly stated by the authors in paragraph 4.1, microglia generate from the yolk sac and invade the brain during neurodevelopment. The necessity to use differentiation protocols toward the mesodermal lineage is simply due to the different embryonic origin of microglia and neuronal cells. Sharing the residence in the human brain with neurons cannot be linked to their distinct developmental origin.

Response:  As the reviewer suggested, this sentence is removed.

Minor comments:

- at line 54 delete “recently” to avoid a repetition.

Response:  As the reviewer suggested, it is deleted.

- at lines 94 delete "cells or" to avoid redundance in the sentence.

Response:  As the reviewer suggested, it is deleted.

- In the introduction some sentences are still difficult to read and are not properly written (i.e. - at lines 91-93). I would recommend to have the manuscript checked by a native English speaker.

Response:  As the reviewer suggested, we revised the sentence in the introduction, which now reads “However, most of recent brain organoids derived from hiPSCs lack several critical components, e.g., vascular cells, microglia, and BBB feature, in the human brain [1].  So fully mimicking the function of human brain has not been achieved.”

In addition, we have the manuscript checked by a native English speaker, and made the minor corrections throughout the whole manuscript.